# Weibull Reliability and Mechanical Properties of Chemical Strengthening Lightweight Glass Containers Using Spray Coating

Kyung Won Min [1,2], Jae Ho Choi [1,2], YoonSung Jung [1,2], Young Min Byun [1,2], Won Bin Im [1] and Hyeong-Jun Kim [2,*]

1   Division of Materials Science and Engineering, Hanyang University, 222, Wangsimni-ro, Seongdong-gu, Seoul 04763, Republic of Korea
2   Engineering Ceramic Center, Korea Institute of Ceramic Engineering and Technology, Icheon 17303, Republic of Korea
\*   Correspondence: goldbud@kicet.re.kr; Tel.: +82-31-645-1480

**Abstract:** In this study, the effects of heat-treatment temperature and hot-end coating (HEC) by spray coating on the mechanical properties and reliability of lightweight glass bottles were investigated. When the chemical strengthening occurred at Tg, the hardness and impact strength increased by 163% and 198%, respectively. All specimens exhibited improved mechanical properties with chemical strengthening, regardless of the HEC. The strengthening effect was relatively large in the absence of HEC. However, the distribution of the impact strength was smaller when HEC was applied. Compared to non-HEC bottles, HEC glass bottles had higher Weibull modulus values after chemical strengthening, which increased their reliability. Therefore, it is possible to chemically strengthen a lightweight glass bottle by spray-coating. Chemically strengthened lightweight glass bottles with excellent mechanical properties and high reliability can be produced when both HEC and chemical strengthening are applied.

**Keywords:** chemical strengthening; lightweight glass bottle; reliability; accelerated life test; Weibull modulus

## 1. Introduction

The use of microplastics is a concern worldwide because of their adverse effects on water and aquatic ecosystems and their harmful effects on humans. Various studies have attempted to address this problem. Among these, studies using glass bottles as an alternative to plastic containers are gaining popularity.

For glass to become an eco-friendly alternative container, it must be lightweight and highly durable, which are advantages of plastics. However, glass bottles are relatively heavy and brittle. Many glass bottle companies have attempted to reduce the weight of bottles by thinning the bottles to make them lighter [1,2]. However, because glass weight reduction has a trade-off relationship with strength, it is necessary to increase its strength.

Hot-end coating (HEC) increases scratch resistance by coating $SnO_2$ or $TiO_2$ on the surface and cold-end coating (CEC) improves the lubricity of bottles by spraying polyethylene on them. Both HEC and CEC have been applied to improve the strength of glass bottles. However, an additional surface strengthening treatment is required to reduce the weight of the glass bottle [3–5].

Surface strengthening includes both thermal and chemical strengthening. Thermal strengthening is used for float glass. Some domestic companies apply it to glass containers. Thermal strengthening caused consumer distrust due to problems such as self-destruction. [6].

Conversely, chemical strengthening is applied using heat treatment by dipping glass into a $KNO_3$ molten salt [7–14]. However, this method is difficult to apply in glass bottle factories which use continuous production processes.

Recently, continuous chemical-strengthening technology has been developed involving spraying chemical strengthening salts onto glass [15,16]. Compared to dipping, the spray method is more suitable for continuous coating, but involves relatively less salt. Therefore, the amount of K ion substitution required to achieve chemical strengthening is expected to decrease. However, few studies have investigated the reinforcing effect when the spray method is applied in glass bottle factories.

This study sought to elucidate the chemical-strengthening effect of applying the spray-coating method to the surface of a lightweight glass bottle by inducing compressive stress. The feasibility of achieving chemical strengthening by applying the spray-coating method in a continuous production process was also investigated by evaluation of the presence or absence of HEC in glass bottles, changes observed in physical properties according to the strengthening temperature, and by statistical analysis.

## 2. Materials and Methods

Figure 1 shows a chemical-strengthening furnace that can perform spray coating. A molten salt container was connected to the sprayer. The spray nozzle is an instrument in which molten salt rises from the molten salt container to the nozzle and is sprayed via the Venturi effect. The glass bottle could be rotated so that the salt could be applied to all sides of the bottle when spraying the molten salt. Therefore, using the chemical-strengthening furnace, the type of molten salt, injection pressure, time, vial rotation speed, and the tempering temperature and time could be controlled. The rotation speed of the glass bottle, the injection pressure, and the injection time were 250 RPM, 3 bar, and 20 s, respectively. The molten salt for strengthening was fixed with $KNO_3$ (99%, DaeJung, Korea).

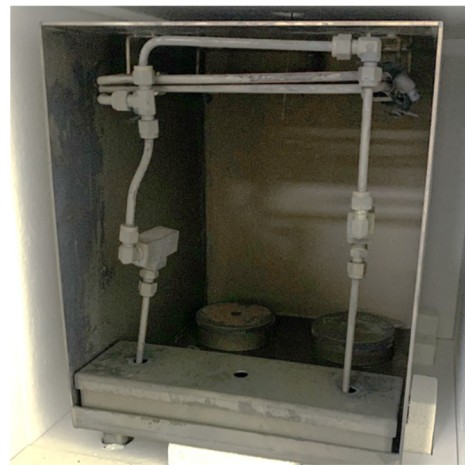

**Figure 1.** Interior image of chemical-strengthening furnace.

The heat-treatment temperature was set to $T_g - 50$, $T_g$, and $T_g + 50$. The heat-treatment holding time was fixed at 1 h in accordance with the annealing time applied in the factory. A total of 20 glass samples were used to evaluate strengthening. In addition, treatment was applied in both the presence and absence of HEC treatment of the glass bottle. HEC-untreated glass bottles were denoted as A and HEC-treated glass bottles as B. A detailed description of the chemically strengthened glass bottles is provided in Table 1; Figure 2 shows a thermal expansion graph.

**Table 1.** Glass-bottle naming according to furnace temperature and HEC.

| | A-X | A-1 | A-2 | A-3 | B-X | B-2 |
|---|---|---|---|---|---|---|
| | | HEC O | | | HEC X | |
| Furnace Temperature (°C) | X | $T_g - 50$ | $T_g$ | $T_g + 50$ | X | $T_g$ |

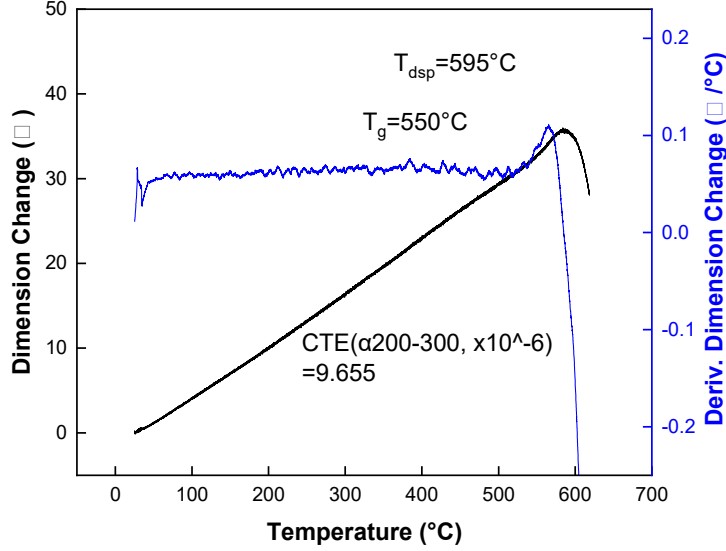

**Figure 2.** Thermal expansion graph of glass bottle.

The glass bottles supplied by Company K had a common soda-lime glass composition. The composition and thermal properties of the glass are listed in Table 2. $T_g$, $T_{dsp}$, and the coefficient of thermal expansion were measured using a thermos mechanical analyzer (Q400, TA Instruments, New Castle, DE, USA).

**Table 2.** Composition and thermal properties of soda-lime glass.

| Composition (wt.%) | |
|---|---|
| $SiO_2$ | 72.64 |
| $Na_2O$ | 13.91 |
| $CaO$ | 10.86 |
| $MgO$ | 0.22 |
| $Al_2O_3$ | 1.8 |
| $K_2O$ | 0.43 |
| $Fe_2O_3$ | 0.04 |
| $BaO$ | 0.1 |
| $Cr_2O_3$ | 0 |
| **CTE($\alpha_{200-300}$, $\times 10^{-6}$)** | 9.655 |
| **$T_g$(°C)** | 550 |
| **$T_{dsp}$ (°C)** | 595 |

The Weibull distribution can be used to estimate important life characteristics of a product, such as its reliability or probability of failure and the mean life. After measuring the impact strength, the reliability of the glass bottles, with and without strengthening and HEC treatment, was evaluated using Weibull analysis. Minitab Statistical Software version 2016 was used for the analysis.

## 3. Results and Discussion

A study was conducted based on the following questions regarding the changes in mechanical properties when glass bottles were chemically strengthened by spraying: (1) Does

strengthening occur even if there are insufficient chemical strengthening salts and reinforcement time? (2) If strengthening occurs, what is the required temperature within a limited time period? (3) Does HEC affect the reinforcement? (4) Does chemical strengthening improve reliability? Chemical strengthening was achieved despite the lack of chemical strengthening salt amount and time.Both the hardness and impact strength improved. Figures 3 and 4 show the changes in hardness and impact strength values observed according to the chemical-strengthening temperature. Both A-series without HEC treatment and B-series with treatment showed maximum hardness and impact strength when heat treated at Tg. The A-series samples chemically strengthened at a high temperature of Tg + 50 decreased both hardness and impact strength.

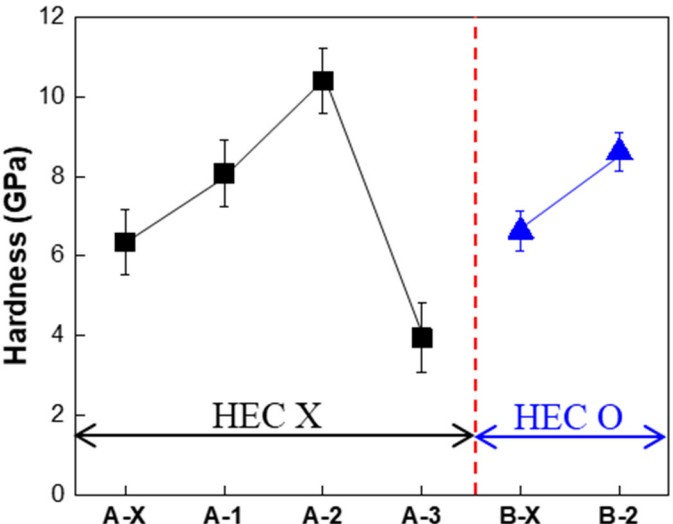

**Figure 3.** Vickers hardness of glass bottle according to furnace temperature and HEC.

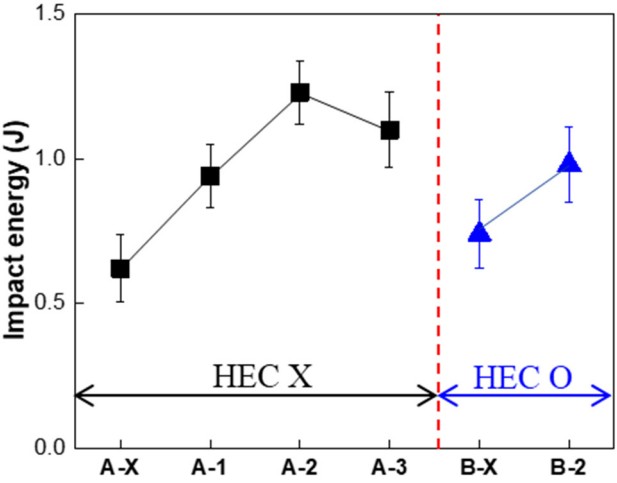

**Figure 4.** Impact energy of glass bottle according to furnace temperature and HEC.

Chemical strengthening of soda-lime glass treated at 100 °C lower than the $T_g$ has been reported [17]. However, in this study, the strengthening time was short, and the quantity of $KNO_3$ salt was considered insufficient. Hence, heat treatment up to a relatively high temperature $T_g$ was necessary to achieve chemical strengthening of the glass bottle. Therefore, the chemical strengthening potential of the spray method and the temperature required were confirmed. The chemical strengthening occurring following spraying at a temperature higher than $T_g$ can be interpreted as a stress relaxation phenomenon resulting from viscous flow [18].

For the hardness and impact strength evaluation without HEC treatment, the two-sample *t*-test *p*-values obtained were 0.022 and 0.011, respectively (smaller than 0.05). This indicated statistically greater hardness and impact strength values than those obtained with HEC treatment. The improvement in the mechanical properties with HEC treatment (B), which was observed to be smaller than that occurring without HEC treatment (A), can be attributed to hindering of the surface K and Na substitution reaction by HEC.

The relationship between the mechanical properties was determined by analyzing the degree of strengthening. Figure 5 shows the compressive stress values observed according to the strengthening temperature applied. As for the hardness and impact strength values, the compressive stress increased until the HEC bottle was heat-treated up to $T_g$. It then decreased when strengthened at $T_g$ + 50. In the samples subjected to HEC treatment (B-2), the compressive stress showed a stress value 50% that of A-2 samples at an equal strengthening temperature. Figure 6a shows the depth of the reinforced layer (DOL) as a function of the tempering temperature. DOL increased up to Tg + 50. In the HEC-treated samples (B-series), there was no statistically significant difference in the average depth of chemical strengthening compared to the A-2 samples tempered at the same Tg temperature, but the relative variance was large. It is thought to mean that the HEC coating was not uniform for each glass sample.

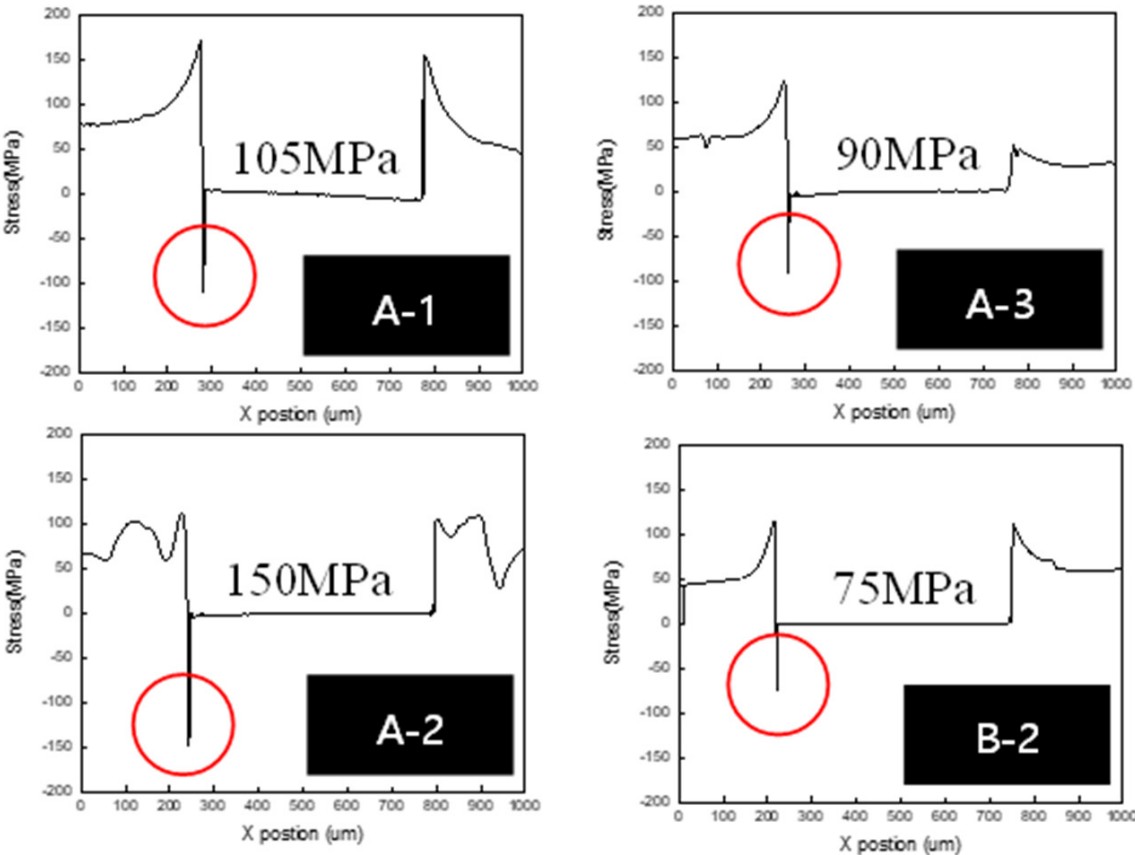

**Figure 5.** Residual stress profile and CS value of chemically strengthened glass bottles.

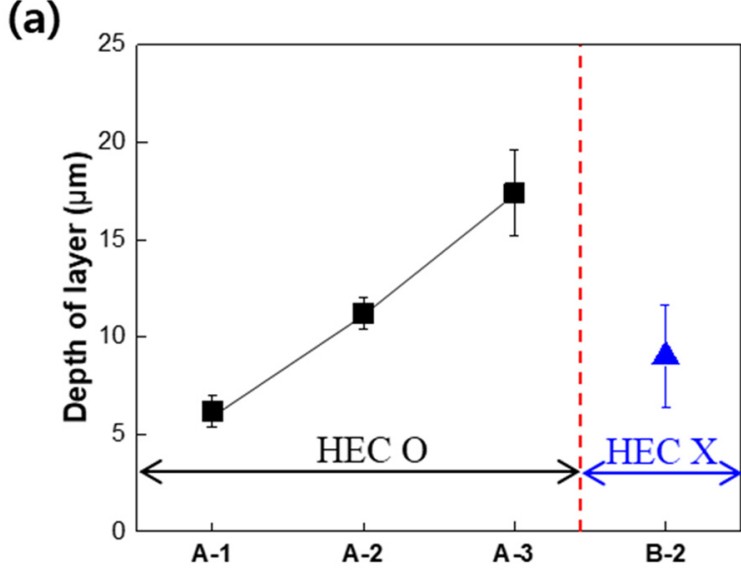

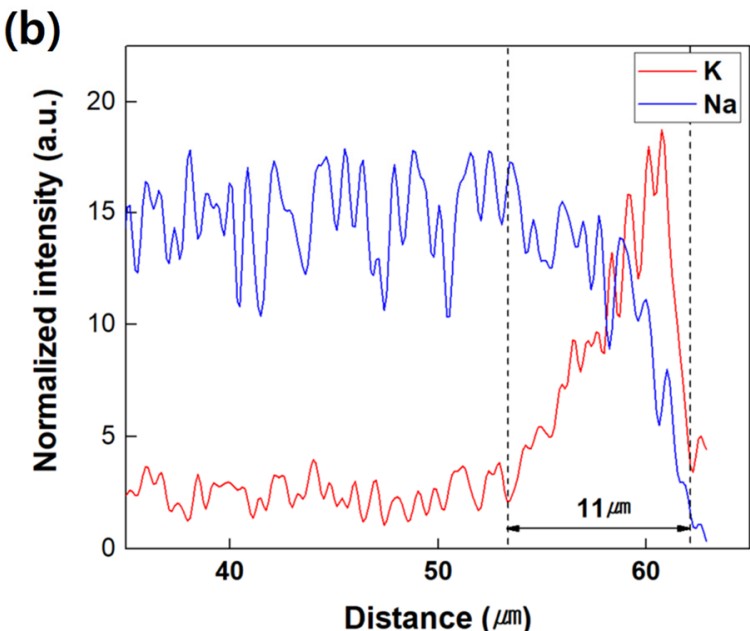

**Figure 6.** DOL of glass bottles according to furnace temperature and HEC (**a**), EDS linescan profile for A-2 (**b**).

Figure 7 shows a Weibull graph of the impact strength according to the heat treatment conditions. Table 3 lists the Weibull modulus (shape parameter). The bottles with HEC treatment (B) had higher Weibull coefficient values than those without HEC treatment (A). All the bottles without HEC treatment (A-series), except those strengthened at $T_g - 50$, had a lower Weibull modulus than those without heat treatment. However, in those with HEC treatment (B-series), the Weibull modulus increased after strengthening.

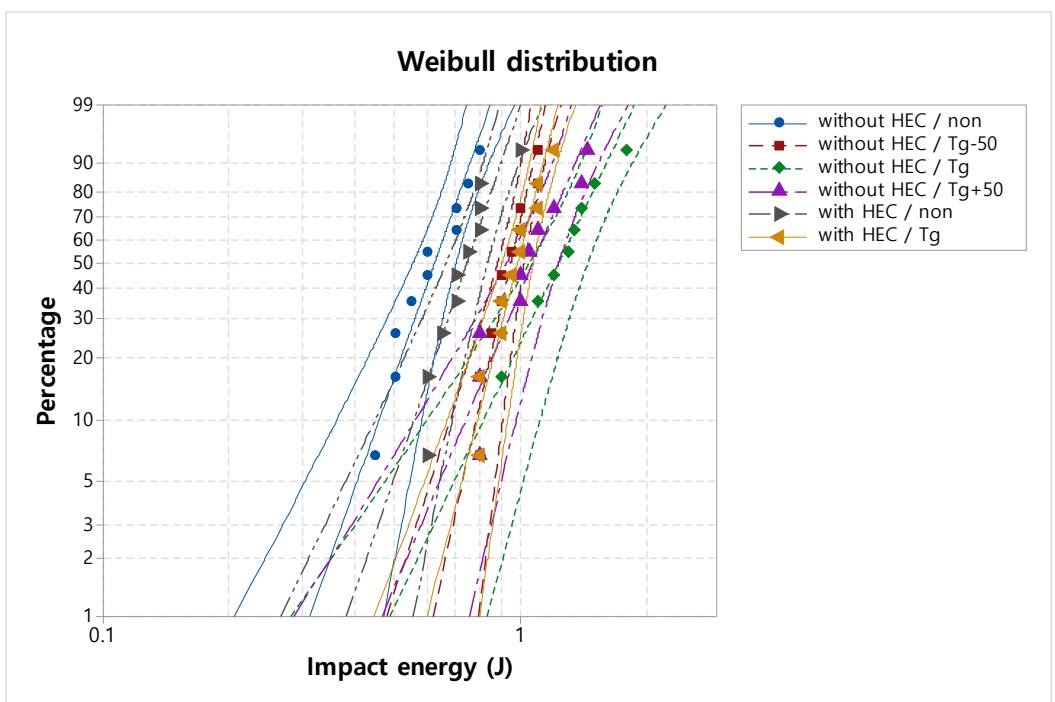

**Figure 7.** Percentage increase as a function of impact energy and Weibull distribution for chemically strengthened bottles.

**Table 3.** Weibull modulus of chemically strengthened glass bottle with HEC.

|  | Weibull Modulus |
|---|---|
| A-X | 6.1318 |
| A-1 | 9.8305 |
| A-2 | 4.5340 |
| A-3 | 5.0906 |
| B-X | 6.3857 |
| B-2 | 8.4670 |

The Weibull modulus is the slope of the Weibull distribution graph, which indicates the reliability [19–21]. Regardless of whether chemical strengthening or HEC was applied, the Weibull coefficients all exceeded one. The failure rate increased according to the amount of impact. Those exceeding 3.5 had an approximately normal probability distribution. This implies that gradual failure occurs in response to changes in the impact strength.

The A-2 and B-2 samples were subject to different HEC treatments. The A-2 samples had higher average strength, but the Weibull modulus value was small. This means that A-2 showed greater strength, but that the bottle breakage probability distribution was relatively wide. From a long-term perspective, the application of both HEC and chemical strengthening at the same time is recommended as a more effective method for strengthening glass containers.

## 4. Conclusions

In this study, the effects of heat treatment temperature and HEC treatment on the mechanical properties of glass were investigated by chemically strengthening glass bottles with spray coating.

When a spray-coating method with less K substitution for diffusion was used compared to a dipping method, the glass hardness and impact strength were improved by up to 163% and 198%, respectively, even if the samples were strengthened over a short

period. The mechanical properties of the glass were improved by chemical strengthening irrespective of HEC coating. The effect of chemical strengthening was relatively large in the absence of HEC. However, the Weibull modulus was higher in the distribution of impact strength for glass bottles to which HEC was applied. Therefore, it is necessary to apply both HEC and chemical strengthening to create a lightweight glass bottle with excellent mechanical properties and high reliability.

The results of this study suggest that spray-based chemical strengthening can be used in conventional manufacturing processes. This means that strength enhancement in lightweight bottles can be achieved through chemical strengthening without adversely affecting continuous glass bottle production. However, it is also necessary to consider how to ensure that the surfaces of all containers are chemically and uniformly hardened. Since there are also returnable bottles, we plan to review whether chemical reinforcement exerts its effect even when reused.

**Author Contributions:** Conceptualization, methodology, formal analysis and writing—original draft, K.W.M. and H.-J.K.; data curation, Y.J. and Y.M.B.; review and editing, W.B.I. and J.H.C.; project administration and funding acquisition, H.-J.K. All authors have read and agreed to the published version of the manuscript.

**Funding:** This study was supported by the Ministry of Trade, Industry, and Energy (Korea) [20010268]. [Project Name: Development of super-lightweight high-strength glass manufacturing technology to reduce microplasticity].

**Data Availability Statement:** Not applicable.

**Conflicts of Interest:** The authors declare no conflict of interest.

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
