# Peer review of "Weibull Reliability and Mechanical Properties of Chemical Strengthening Lightweight Glass Containers Using Spray Coating"

_processes, doi:10.3390/pr11010015_

Round 1

Reviewer 1 Report

Authors presented the effects of the heat treatment temperature and HEC on the mechanical properties and chemically strengthening of glass bottles with spray coating. This manuscript is interesting and can be accepted for publication after minor revision taking into account next point:

Further characterization of the spray coated films should be carried out in order to understand the properties of the films. For example, potassium and sodium concentration profile should be investigated by EDX on the fracture surface for studying their concentration on the glass surface before and after the ion exchange process.

Spray coating technique is a useful method for this purpose at industrial level. However, this work dos not compare the advantages of the employed methodology with other ones, such as dipping, since the amount of ion exchange is expected to decrease with spray deposition method. Thus, the results of a sample coated by another technique should be added and discussed, focusing in the amount of K/Na substitution and the mechanical properties of the resulting coated sample. Moreover, the results reported in the literature for related materials should also be incorporated for comparison purpose.

Author Response

Thank you for reviewing my thesis.
Below is my response to reviewer 1 comment.

Further characterization of the spray coated films should be carried out in order to understand the properties of the films. For example, potassium and sodium concentration profile should be investigated by EDX on the fracture surface for studying their concentration on the glass surface before and after the ion exchange process.

--> DOL was measured through EDX and linescan, and the representative linescan profile of A-2 was added to figure.6.

Spray coating technique is a useful method for this purpose at industrial level. However, this work dos not compare the advantages of the employed methodology with other ones, such as dipping, since the amount of ion exchange is expected to decrease with spray deposition method. Thus, the results of a sample coated by another technique should be added and discussed, focusing in the amount of K/Na substitution and the mechanical properties of the resulting coated sample. Moreover, the results reported in the literature for related materials should also be incorporated for comparison purpose.

--> Of course, dipping can be further strengthened, but it was not tested because it was not suitable for continuous process application, which is the method of glass bottle manufacturing plants, and that is the purpose of this study. However, I will proceed with the comparison tests if the reviewer deems it necessary. Studies have experimented though the dipping method and fragments from broken glass bottles. The study conducted in the spray method tested with the form of a float, not a glass bottle. However, in my study, the glass bottle did not break and was strengthened with a spray coating. Therefore, in this study, glass bottle spray coating chemical strengthening doesn't have references.

Reviewer 2 Report

In this manuscript, Kyung et al reported the weibull reliability and mechanical properties of chemical strengthened lightweight glass containers using spray coating, the paper can be accepted after the following issue were concerned.

1. The surface property of the sample should be checked vs SEM.

2. What about the surface roughness of the spray coating process?

3. There are several failure reasons for the spray coating, what is the main reason here. Can the authors provide solid evidents for that?

4. Laterest progress of this field can be added. e.g. Materials Today Physics, 2022, 29, 100919

Author Response

Thank you for reviewing my article.
Below is my response to reviewer 2 comment.

1. The surface property of the sample should be checked vs SEM.

--> As shown in the attached file, there is no change in the surface before and after chemical strengthening.

2. What about the surface roughness of the spray coating process?

--> As in the attached file, there is no difference in surface roughness before and after chemical strengthening.

3. There are several failure reasons for the spray coating, what is the main reason here. Can the authors provide solid evidents for that?

--> Unlike dipping, which is a process previously used, spray coating may have a problem in that strengthening salt does flow down and a small amount of strengthening salt is coated to it so that strengthening does not occur. But that problem was solved because the glass bottle was chemically strengthened. This study is solid evidence.

4. Laterest progress of this field can be added. e.g. Materials Today Physics, 2022, 29, 100919

--> Recent related studies are references 13 and 14. However, my research is the latest and only research because the existing research has not been conducted by chemically strengthening the glass bottle by spray coating.

Round 2

Reviewer 2 Report

The manuscript has been improved and it can now be accepted as it is.